# Nuclear Retention of mRNAs Through Paraspeckle Protein Binding to a Sequence Determinant in 3′UTR

**DOI:** 10.3390/ijms26136488

**Published:** 2025-07-05

**Authors:** Audrey Jacq, Denis Becquet, Bénédicte Boyer, Séverine Guillen, Maria-Montserrat Bello-Goutierrez, Marie-Pierre Blanchard, Claude Villard, Maya Belghazi, Manon Torres, Jean-Louis Franc, Anne-Marie François-Bellan

**Affiliations:** 1INP, AMU-CNRS UMR 7051, Faculté des Sciences Médicales et Paramédicales, Campus Timone, 27 Bd Jean Moulin, 13385 Marseille CEDEX 05, France; ajacq@tamu.edu (A.J.); denis.becquet@univ-amu.fr (D.B.); benedicte.boyer@univ-amu.fr (B.B.); severine.guillen@univ-amu.fr (S.G.); maria-montserrat.goutierrez@univ-amu.fr (M.-M.B.-G.); marie-pierre.blanchard@igh.cnrs.fr (M.-P.B.); claude.villard@univ-amu.fr (C.V.); maya.belghazi@univ-amu.fr (M.B.); manon.torres@charite.de (M.T.);; 2Biology Department, Texas A&M University, BSBW354, 424 Nagle Street, 3258, College Station, TX 77843-3258, USA; 3Montpellier Ressources Imagerie, BioCampus, University of Montpellier, CNRS, INSERM, 34396 Montpellier CEDEX 05, France; 4Institut de Microbiologie de la Méditerranée (IMM), CNRS, Aix-Marseille Université, 13009 Marseille, France; 5Charité–Universitätsmedizin Berlin Campus Charité Mitte Chronobiology in the Charité CrossOver (CCO), Virchowweg 6 Charitéplatz 1, 10117 Berlin, Germany

**Keywords:** paraspeckles, mRNA targets, RNA nuclear retention, RNA-binding protein, 3′UTR sequence determinant

## Abstract

Paraspeckles are nuclear membraneless structures composed of a long non-coding RNA, Nuclear-Enriched-Abundant-Transcript-1, and RNA-binding proteins, which associate with numerous mRNAs. It is therefore believed that their cellular function is to sequester in the nucleus their associated proteins and/or target mRNAs. However, little is known about the molecular determinant in mRNA targets that allows their association to paraspeckles, except that inverted repeats of Alu sequences (IRAlu) present in the 3′UTR of mRNAs may allow this association. While in a previous study we established the list of paraspeckle target RNAs in a rat pituitary cell line, we did not find, however, inverted repeated SINEs, the rat equivalent of primate IRAlus in 3′UTR of these RNAs. By developing a candidate gene strategy, we selected a paraspeckle target gene, namely calreticulin mRNA, and we searched for other potential RNA recruitment elements in its 3′UTR, since 3′UTRs usually contain the sequence recognition for nuclear localization. We found a 15-nucleotide sequence surrounded in 5′ by a C-rich sequence, which is present as a tandem repeat in the 3′UTR of this mRNA and which is involved in the nuclear retention by paraspeckles. As shown by mass spectrometry analysis, 6 proteins bound to the 15-nucleotide sequence are paraspeckle proteins and constitute, therefore, bridging proteins between paraspeckles and target mRNAs.

## 1. Introduction

The paraspeckles, discovered in 2002 [1], are nuclear bodies found in almost all of the cultured cell lines and primary cultures from tissues [2], except for embryonic stem cells [3]. They are usually detected as a variable number of discrete dots found in close proximity to nuclear speckles [2]. The structural element of paraspeckles is a long non-coding RNA, nuclear-enriched abundant transcript one (Neat1), which exists under a short and a long transcript generated from the same promoter, previously identified as MENε (Neat1_1) and MENβ (Neat1_2), respectively [4,5]. The long isoform is essential for the formation and maintenance of paraspeckles. More than 60 proteins have been identified thus far to be associated with paraspeckles [6], among which 4 RNA-binding proteins, including 3 members of the Drosophila Melanogaster Behavior Human Splicing (DBHS) family proteins (NONO, PSPC1, and SFPQ) and RNA-binding motif protein 14 (RBM14), are considered as core paraspeckle proteins [2,7,8]. Another crucial paraspeckle protein, the heterogeneous nuclear ribonucleoprotein K (HNRNPK), appeared to affect the production of the essential NEAT1_2 isoform by negatively regulating the 3′-end poly-adenylation of the NEAT1_1 short isoform [9]. Paraspeckles have also been shown to associate with numerous mRNAs. It is therefore believed that the cellular function of paraspeckles is to sequester in the nucleus their associated proteins and/or their target mRNAs. In a rat pituitary cell line, the GH4C1 cells, we identified using a Neat1 RNA pull-down procedure [10] followed by RNA sequencing, 3928 RNAs that are paraspeckle’s targets [11]. In these cells, we have shown that the expression of all major components of paraspeckles, including Neat1 and core paraspeckle proteins, followed a circadian pattern that leads to rhythmic variations in paraspeckle number within these cells [11]. Thanks to their circadian expression pattern and given their functions in mRNA nuclear retention, it was shown that paraspeckles rhythmically retain target RNAs in the nucleus of the cells and that this rhythmic nuclear retention leads to the rhythmic expression of the corresponding genes [11,12]. However, the way by which mRNAs are bound to paraspeckles is largely unknown. In particular, the molecular sequence determinants that enable mRNAs to be recognized and targeted by paraspeckles are poorly understood even though it has been shown that inverted repeats of Alu sequences (IRAlu) present in the 3′UTR of mRNAs enable these mRNAs to associate with paraspeckles [3,8,13]. Accordingly, we and others have shown that insertion of IRAlus in the 3′-UTR of an Egfp reporter gene induced both nuclear retention of Egfp mRNA and reduction in EGFP protein expression in the cytoplasm [11,13]. In addition, endogenous mRNAs containing IRAlus in their 3′-UTRs, such as Nicolin 1 (NICN1) or Lin 28, have been shown to be retained in the nucleus by paraspeckles [13,14], providing conclusive evidence that the presence of IRAlus in 3′-UTRs of genes allows paraspeckles to sequester mature mRNAs within the nucleus.

In human cells, hundreds of genes contain inverted repeated short interspersed nuclear elements (SINEs) (mainly Alu elements) in their 3′-UTRs. Alu elements are unique to primates, accounting for almost all of the human SINEs and over 10% of the genome. Their abundance leads to the frequent occurrence of IRAlus in gene regions. Unexpectedly, however, in the 3′-UTR of the 3928 RNAs we identified as Neat1 targets in rat GH4C1 cells, we found no inverted SINE repeats, the rat equivalent of primate IRAlus (unpublished results). The question of whether sequence determinants other than IRAlus in the 3′UTR of mRNAs enable them to be paraspeckle’s targets therefore remains unanswered. To identify potential RNA recruitment elements, we chose to adopt a candidate gene strategy based on analysis of the ability of the 3′-UTR candidate gene to associate with paraspeckles. To this end, we selected a candidate gene, namely calreticulin (Calr) mRNA, previously shown to be both a post-transcriptional cycling transcript in the liver [15] and a target of paraspeckle, whose circadian nuclear retention depended on Neat1 in our cell line [11]. We showed that the association of Calr mRNA with paraspeckles involved its 3′UTR, which contained a 15-nucleotide sequence found as a tandem repeat. This sequence, flanked in 5′ by a stretch of C, was shown to be involved in the nuclear retention of the 3′UTR of Calr mRNA by paraspeckles and was further shown to specifically bind several paraspeckle proteins that probably constitute bridging proteins between paraspeckles and target mRNAs.

## 2. Results

### 2.1. Post-Transcriptional Calr mRNA Circadian Expression

According to Menet et al. [15] and Beckwith and Yanovsky [16], the rhythmic pattern of mRNAs is believed to be regulated at the post-transcriptional or transcriptional level depending on whether their corresponding pre-mRNA is arrhythmic or rhythmic (Figure 1A, Right Panel). While Calr mRNA displayed a circadian expression pattern in GH4C1 cells, Calr pre-mRNA did not (Figure 1A). Experimental Calr mRNA values could indeed be fitted by the following equations: Calr: y = 111.8 – 21.35 × sin (0.261x + 0.938) with an R^2^ = 0.688. By contrast, Calr pre-mRNA levels could not be significantly fitted by a sine wave equation with a period of 24 h (R^2^ < 0.55) (Figure 1A). It then appeared that Calr pre-mRNA did not display a rhythmic expression pattern (Figure 1A), despite high fluctuations of Calr pre-mRNA levels during the time course, as previously shown to occur for 80% of pre-mRNA of post-transcriptionally regulated mouse liver cycling transcripts [15]. Accordingly, it appeared that in GH4C1 cells, the rhythmic pattern of Calr mRNA was regulated at a post-transcriptional level. An example of a rhythmic mRNA with circadian transcription was provided with Rpa1 (replication protein A1), whose values of both mRNA and pre-mRNA could be fitted by non-linear sine wave equations with a period of 24 h (Rpa1 pre-mRNA: y = 113.2 + 21.3 × sin (0.261x – 1.377) with an R^2^ = 0.563 and Rpa1mRNA: y = 120.1 – 20.7 × sin (0.261x + 0.287) with an R^2^ = 0.617) (Figure 1A).

It then may be concluded that, as expected for an mRNA paraspeckle target, the control of the Calr circadian expression pattern took place at a post-transcriptional level, and this was not the case for Rpa1, which we did not find to be a paraspeckle target in our previous study [11].

### 2.2. Association of Calr mRNA with Paraspeckles in GH4C1 Cells

To provide evidence for the association of Calr mRNA with paraspeckles, we investigated its association with all major components of the nuclear bodies, namely Neat1 and the four major core paraspeckle proteins.

The association of Calr mRNA with Neat1 was investigated using the Neat1 RNA pull-down procedure we previously described [10,11]. In agreement with our previous reports [10,11], the procedure was shown here to allow a 450 to 600 higher enrichment in Neat1 RNA by using specific oligonucleotides (SO1 and SO2) compared to non-specific oligonucleotides (NSO) probes (Appendix A). Furthermore, in these experiments, Calr mRNA was shown to be significantly enriched relative to input when SO probes directed against Neat1 were used as compared to a NSO probe, showing clearly that Calr mRNA was associated with Neat1 (Figure 1B). By contrast, there was no enrichment relative to input in Rpa1 mRNA after use of the two SO probes as compared to NSO (Figure 1B), showing that Rpa1 was not associated with Neat1, in agreement with our previous results [11]. While Calr mRNA enrichment was statistically significant with the two SOs used (SO1 and SO2) compared to NSO, SO2 induced a significantly higher enrichment (Figure 1B); this prompted us to use this oligonucleotide probe in our further experiments.

To investigate the association of Calr mRNA with paraspeckle proteins, we used RNA-immunoprecipitation (RIP) experiments with antibodies directed against the four core paraspeckle proteins, NONO, SFPQ, RBM14, or PSPC1. A 3- to 8.5-fold enrichment in Calr mRNA was obtained with these antibodies as compared to an irrelevant non-specific antibody (Figure 1C). By contrast, in these experiments, no enrichment in Rpa1 mRNA was obtained with the four specific antibodies, showing that Rpa1 mRNA was not associated with core paraspeckle proteins (Figure 1C). The association of Calr mRNA with the four core protein components of paraspeckles reinforced the view that Calr mRNA bound paraspeckles.

Finally, we performed dual FISH experiments to provide additional morphological indication that Calr mRNA may be associated with paraspeckles visualized by Neat1 RNA staining (Appendix A). As already described, Neat1 RNA staining under a confocal laser scanning microscope appeared as regular punctates within the boundaries of the nucleus (Appendix A) [11]. By contrast, Calr mRNA staining was mainly localized in the cytoplasm, while some punctates also appeared in the nucleus (Appendix A). Among these nuclear punctates, some merged with Neat1 RNA staining (Appendix A), showing that Calr mRNA could be associated with Neat1 RNA. By contrast, some Neat1 RNA stained punctate did not merge with Calr mRNA staining (Appendix A).

Taken together, all these results supported the idea that Calr mRNA was closely associated with paraspeckles in GH4C1 cells and validated its choice as a gene candidate to search for the molecular mechanism of this association.

### 2.3. Involvement of 3′UTR in the Binding of Calr mRNA to Paraspeckles

Paraspeckles are known to retain in the nucleus RNAs containing duplex structures from inverted repeats of the conserved Alu sequences (IRAlu elements) within their 3′-UTR, as shown for the Nicolin 1 (NICNI) gene [14]. We used previously generated GH4C1 cell lines stably transfected by constructs in which an IRAlu sequence from the 3′-UTR of the NICN1 gene or an Alu element as a control was inserted each between the EGFP cDNA 3′-UTR region and the SV40 polyadenylation signal of the expression vector pEGFP-C1 [11]. To evaluate whether the 3′UTR of Calr mRNA could bind paraspeckle nuclear bodies, the same strategy was used, and the 3′UTR of Calr mRNA was cloned by PCR, inserted in the pEGFP-C1 vector, and the construct was stably transfected into GH4C1 cells. Neat1 RNA pull-down experiments were performed in the three cell lines containing constructs with either Alu- or IRAlu elements or the 3′UTR of Calr mRNA. Neat1 enrichment obtained with SO2 was 300 to 450, significantly higher compared to NSO but not different between the three cell lines (Appendix A). Consistent with the known ability of IRAlu elements to bind paraspeckles, we found, as previously reported [11], that the amounts of Egfp mRNA retrieved after Neat1 RNA pull-down by SO2 compared to NSO were 2-fold significantly higher in IRAlu compared to the Alu-egfp cell line (Figure 2A). The efficacy of the 3′UTR of Calr mRNA to enrich Egfp mRNA after Neat1 RNA pull-down with SO2 compared to NSO was then compared to that of Alu- or IRAlu-elements. As shown in Figure 2A, Egfp enrichment after Neat1 RNA pull-down was significantly higher in the cell line expressing 3′UTR-Calr as compared to the cell line expressing Alu-containing Egfp mRNA, and no statistical difference was obtained between cell lines expressing either 3′UTR-Calr- or IRAlu-containing Egfp mRNA (Figure 2A). The 3′UTR of Calr mRNA associated with Neat1 as efficiently as the IRAlu element.

We then performed nuclear and cytoplasmic separation to investigate the influence of 3′UTR Calr mRNA on the nucleo-cytoplasmic distribution of Egfp reporter mRNA. Cross-contamination between nuclear and cytoplasmic fractions was ruled out by showing that a nuclear transcription factor, ATF2, and a cytoplasmic protein, MEK1/2, were expressed exclusively in nuclear and cytoplasmic fractions, respectively [17] (Appendix A). We showed that both the IRAlu from Nicn1 and the 3′UTR of Calr mRNA caused a significantly greater nuclear retention of the Egfp mRNA when compared with the corresponding Alu element, with no significant difference between the 3′UTR-Calr and IRAlu element (Figure 2B). This clearly showed that the 3′UTR-Calr sequence retained Egfp mRNA in the nucleus as efficiently as the IRAlu sequence did. Taken together, these results were consistent with the involvement of the 3′UTR of Calr mRNA in the binding of Calr mRNA to paraspeckle nuclear bodies.

### 2.4. Delineation of Sub-Regions in 3′UTR-Calr mRNA Engaged in Paraspeckle Binding

To further characterize the sub-regions of the 3′UTR of Calr mRNA that are involved in the binding to paraspeckles, three fragments (3′UTR-C1 (166 nt), 3′UTR-C2 (210 nt), and 3′UTR-C3 (184 nt)) were generated by PCR from the entire 3′UTR of Calr mRNA (Figure 2C). Each of the three fragments was inserted between the Egfp cDNA 3′-UTR region and the SV40 polyadenylation signal of the expression vector pEGFP-C1 to generate constructs that were then stably transfected into GH4C1 cells. Neat1 RNA pull-down experiments were performed in the cell lines containing constructs with 3′UTR Calr mRNA, 3′UTR-C1, 3′UTR-C2, or 3′UTR-C3. In all these cell lines, Neat1 enrichment obtained with SO2 was 300 to 500 significantly higher compared to NSO, but there was no statistical difference between the different cell lines (Appendix A). The efficacy of each fragment was then compared to that of the entire 3′UTR of Calr mRNA to enrich Egfp mRNA after Neat1 RNA pull-down with SO2 versus NSO. Egfp enrichment obtained with 3′UTR-C1 and 3‘UTR-C3 fragments did not differ from that obtained with the entire 3′UTR-Calr; by contrast, Egfp enrichment obtained with 3′UTR-C2 was reduced and significantly lower than that obtained with 3′UTR-C1 (Figure 2D). All three fragments were, however, less able than the entire 3′UTR of Calr mRNA to retain Egfp mRNA in the nucleus; while 3′UTR-C1 and 3′UTR-C3 induced a same retention of about 30% compared to the entire 3′UTR, 3′UTR-C2 was significantly less efficient, inducing a 20% nuclear retention compared to that obtained with the entire 3′UTR (Figure 2E). It then appeared that while 3′UTR-C1 and 3′UTR-C3 were as efficient as the entire 3′UTR-Calr to induce the binding of Egfp mRNA to paraspeckles, they were not as efficient as the entire 3′UTR-Calr to retain Egfp mRNA in the nucleus. In addition, 3′UTR-C2 was less efficient both to bind the paraspeckles and to retain Egfp mRNA in the nucleus.

### 2.5. Involvement of a Tandem Sequence of 15 Nucleotides in 3′UTR-Calr mRNA in the Binding to Paraspeckles

Through alignment of the oligonucleotide sequences of the fragments C1 and C3 using the Biostring pairwise Alignment function (Smith-Waterman local alignment), a common sequence of 15 bases was identified in these two fragments, with a T or C (Y) in the 5th position and a G or A (R) in the 11th position (Figure 2F). The 3′UTR of Calr mRNA exhibited then a tandem of the 15-nucleotide sequence (Appendix A). To investigate the involvement of these 15-nucleotide sequences in paraspeckle binding, mutations were introduced in the sequence present in position 37–51 (3′UTR-Mut1, Appendix A), or in position 466–480 (3′UTR-Mut2, Appendix A), or in both positions (3′UTR-Mut1/2) by replacement of every odd base by its complementary inverse base. The 3′UTR of Calr mRNA with Mut1, Mut2, or Mut1/2 was inserted in the pEGFP-C1 vector, and the different constructs were stably transfected into GH4C1 cells. After Neat1 RNA pull-down experiments in all cell lines, Neat1 enrichment obtained with SO2 was 300 to 500 significantly higher compared to NSO but not different between the different cell lines (Appendix A). Enrichment in Egfp mRNA obtained after these mutations was compared to that obtained with the native entire 3′UTR-Calr mRNA. Mutations in the sequence present in position 37–51 as well as mutations in the sequence present in position 466–480 significantly decreased Egfp mRNA enrichment (Figure 2G); however, the dual mutation of the two sequences did not amplify the reduction in Egfp mRNA enrichment (Figure 2G). While Mut1 and Mut2 significantly reduced the relative ratio of nuclear versus cytoplasmic Egfp mRNA distribution when compared to the native entire 3′UTR-Calr, dual mutation of the two sequences did not display an additive or synergic effect (Figure 2H).

### 2.6. Mass Spectrometry Analysis of the Proteins Bounded to the 15 Nucleotides Sequence and Its 5′ Flanked Region

In addition to the 15-nucleotide sequence mentioned above, the alignment of C1 and C3 allowed us to show an enrichment in C in the 6 bases located in 5′ of the 15-nucleotide sequence being ACCACC in C1 and CTCCCT in C3 (Appendix A). The 15-nucleotide sequences found in tandem in the 3′-UTR of Calr mRNA were then surrounded in 5′ by a stretch of C.

We used a 30-base oligonucleotide encompassing the 15-nucleotide sequence from C3 surrounded by 7 nucleotides in 5′ (containing the enriched CTCCCT) and by 8 nucleotides in 3′ (30S, Appendix A). This 30S oligonucleotide was inserted in the pEGFP-C1 vector, and the construct was stably transfected into GH4C1 cells. This 30S oligonucleotide was shown to be as efficient as 3′-UTR-C3 to retain Egfp mRNA within the nucleus (Figure 3A). In contrast, replacement of every odd base by its complementary inverse base in the 15-nucleotide sequence and disruption by mutation of the stretch of C in the 4 adjacent bases in 5′ (30NS) deeply reduced the nuclear retention of Egfp mRNA (Figure 3A).

To identify proteins able to bind the different sequences in the 30S oligonucleotide, RNA-protein pull-down was performed using nuclear protein extracts from GH4C1 cells and different biotinylated 30-bases oligonucleotides: the biotinylated 30S RNA probe corresponding to the 15-nucleotide sequence present in C3 in the 3′UTR-Calr mRNA surrounded in 5′ by 7 nucleotides and in 3′ by 8 nucleotides, a 30S probe with disruption by mutation of the stretch of C in the 4 adjacent bases in 5′ (30SSM) and a non-specific probe (30NS) corresponding to the 30S that included the replacement of every odd base by its complementary inverse base in the 15-nucleotide sequence and disruption by mutation of the stretch of C in the 4adjacent bases in 5′ (Appendix A). Three replicates with each probe were performed, and the experiment was reproduced twice.

Label free mass spectrometry analysis gave a list of 11 proteins that were identified using the 30S and 30SSM probes and not with the 30NS probe (Figure 3B). These 11 proteins (ALF-C1, EWSR1, FUS, HNRNPA0, HNRNPA1, HNRNPA2B1, HNRNPAB, HNRNPD, HNRNPK, PTBP1, and YBX3) were considered as preferentially bound to the 15-nucleotide sequence. Five proteins (DCD, DSP, ILF2, JUP, and NCL) not recovered with the 30NS probe were identified with the 30S but not with the 30SSM, suggesting that these 5 proteins were merely associated with the stretch of C present in 5′ of the 15-nucleotide sequence (Figure 3B). None of these 5 proteins was previously described as a paraspeckle protein component [9]. By contrast, 6 proteins (EWSR1, FUS, HNRNPA1, HNRNPA2B1, HNRNPK, and PTBP1) among the 11 proteins preferentially associated with the 15-nucleotide sequence were included in the list of paraspeckle proteins [9]. It then appeared that paraspeckle proteins were mostly found on the 15-nucleotide sequence where they represent 6/11 (54%) proteins (Figure 3B).

### 2.7. Number of Occurrences of the 15-Nucleotide Sequence Surrounded in 5′ by a C Stretch in the 3′UTR of the Neat1 mRNA Targets

The prevalence of the 15-nucleotide sequence identified in the 3′UTR of the Calr mRNA was determined in the 3′UTR of the 3928 Neat1 RNA targets we previously established [11]. Since the same mRNA can display different lengths of 3′UTR, the list of the Neat1 RNA targets was filtered to retain for each mRNA only the longest 3′UTR referenced. This led to a list of 3347 3′UTR. For this analysis, four mismatches were allowed in the 15-nucleotide sequence. In the list of 3347 3′UTR, we found 2190 times this specific sequence included in 1330 mRNA (Table 1). It then appeared that nearly 40% of the Neat1 RNA targets had at least one identified 15-nucleotide sequence in their 3′UTR, with a mean of 1.65 sequences per gene. After Panther analysis, these Neat1 RNA targets with at least one 15-nucleotide sequence were shown to be functionally associated with chromatin organization, intracellular transport, establishment of localization in the cell, and regulation of gene expression (Appendix A). When considering the background defined as total rat RNAs, of the 19,290 3′UTR retained for the analysis (after filtering for the longest 3′UTR of a single mRNA), 10,575 sequences were identified and found to correspond to 6386 mRNA; near 33% of total RNAs displayed at least one identified 15-nucleotide sequence in their 3′UTR (Table 1). The 15-nucleotide sequence we described in the present paper was then slightly but significantly enriched in the 3′UTR of mRNA that are paraspeckle’s target (X-squared = 55.537, df = 1, *p*-value = 9.17 × 10^−14^). It could then be proposed that the 15-nucleotide sequence described here contributed to paraspeckle binding.

The important prevalence of this 15-nucleotide sequence in total RNAs prompted us to investigate further its relevance. Since in the 15-nucleotide sequence the 5th and the 11th positions were Y and R, respectively, there were 4 variants of this sequence, each with its own number of occurrences. The mean value of those occurrences for the 4 variants was 763 times in the list of the Neat1 RNA targets and 3209 times in total rat RNAs (Appendix A). To evaluate the relevance of these values, we compared them to the number of occurrences of 30 randomly generated sequences of 15 nucleotides with Y or R in the 5th and 11th positions, respectively (30 was the minimum size for a statistically relevant sampling for the global population). The mean value of the number of occurrences for the 4 variants of the 30 random sequences was shown to be significantly lower, 363 times and 1608 times in the list of the Neat1 RNA targets and in the total rat RNAs, respectively (Appendix A). It then appeared that the 15-nucleotide sequence we have identified was about twice as frequent as a random sequence of the same length in the 3′ UTR of both Neat1 RNA targets and total RNAs. Since the prevalence of the 15-nucleotide sequence was statistically relevant, a table of consensus matrix probabilities was therefore built.

The table of consensus matrix probabilities built from the 2190 sequences found in the 3′UTR of Neat1 mRNA targets (Appendix A) allowed the design of a sequence logo (Bioconductor::seqLogo v.1.56.0) (Figure 3C). The frequency matrix and its graphic representation clearly show that there was a core motif of YCCAGGR (Figure 3C, Appendix A).

## 3. Discussion

Although the physiological functions of paraspeckles remain only partly understood, two known functions of paraspeckles are the sequestration of specific transcription factors and/or RNA-binding proteins and the regulation of the expression of specific transcripts via their retention in the nucleus [18]. Thus, Neat1 can regulate expression of genes transcriptionally by sequestering proteins involved in the regulation of gene promoters and co-transcriptionally by binding to mRNAs. While some studies have been undertaken to determine the sequence elements in Neat1 responsible for the recruitment of proteins [19], little is known about potential specific elements in paraspeckle mRNA targets except that some mRNA targets contain inverted repeats derived from SINE repeats in their 3′ UTRs [3,8], an element we did not find in 3′UTRs of paraspeckle mRNA targets in rat GH4C1 cells. Two main strategies can be used to identify recurring sequence patterns or motifs, namely a global computational motif discovery approach or a gene candidate strategy. The first one adopts either traditional and popular methods such as MEME (Multiple Expression motifs for Motif Elicitation) [20] or more recent ones such as Amadeus (A Motif Algorithm for Detecting Enrichment in mUltiple Species) [21] to discover the sequence consensus of RNA-Binding Protein’s binding sites or uses Z-score statistics to search for the overrepresented nucleotides of a certain size. In our case, the main practical limitations of these genome-scale detections of known and/or novel motifs lie in the fact that the known or novel motif has to be overrepresented to be identified and the size of the searched sequence has to be defined. However, a nucleotide sequence may be involved in paraspeckle binding without being overrepresented in paraspeckle mRNA targets, like is the case for IRAlu sequences, and the size of the potential sequence cannot be presupposed a priori. For all these reasons, a gene candidate strategy appeared more suitable to reach our goal.

Among the 3928 paraspeckle RNA targets we reported in GH4C1 cells [11], our choice of gene candidate fell on Calr mRNA, which is shown here to be closely associated not only with Neat1 but also with every major core-protein component of paraspeckles. Furthermore, given that circadian post-transcriptional regulation is shown here to account for its circadian expression pattern as previously shown in the liver [15], and since we previously reported that its circadian nuclear retention is disrupted after Neat1 knock-down [11], it may be assumed that paraspeckle binding plays a role in Calr mRNA expression and circadian pattern [11]. Finally, the 3′UTR of Calr mRNA appears here as efficient as an IRAlu sequence to bind Neat1 and to induce a nuclear retention of a reporter gene, leading to the conclusion that the dissection of 3′UTR Calr mRNA may allow us to uncover sequence elements involved in paraspeckle binding.

The most proximal and the most distal regions of the 3′UTR Calr mRNA identified as two regions able to associate with Neat1 are however far less efficient than the entire 3′UTR at induce a nuclear retention of the reporter gene, suggesting that cooperative effects between different parts of the 3′UTR may be necessary for an efficient nuclear retention of the mRNA. It is tempting to speculate that these cooperative effects between different parts of the 3′UTR may be supported by secondary structures carried by mRNA and as previously suggested, may be determinant for nuclear retention [22]. In any case, the efficiency of the most proximal and the most distal regions of 3′UTR Calr to associate with paraspeckles prompted us to further analyze their molecular features.

Using RNA-RNA interaction prediction algorithms (IntaRNA, http://rna.informatik.uni-freiburg.de/ (accessed on 1 March 2023)) [23], we were able to rule out that the two identified regions of 3′UTR Calr can directly hybridize with Neat1. Indeed, the only part of 3′UTR Calr that offered a significant prediction score for Neat1 interaction is located between the two regions of interest (see Appendix A). While devoid of predictable Neat1 direct interaction sequence, the two regions of interest were shown to display a common sequence of 15 nucleotides involved in paraspeckle binding and Calr mRNA nuclear retention. Interestingly, around 40% of paraspeckle mRNA targets harbor a sequence close to this one in their 3′UTR, with an average of 1.65 sequences per 3′UTR. Furthermore, since the occurrence of this sequence in 3′UTR is slightly overrepresented in mRNA that are paraspeckle targets as compared to all expressed mRNA in the cells, it may be assumed that the 15-nucleotide sequence allows the 3′UTR of mRNA to associate with paraspeckles. However, the high occurrence of the sequence in all expressed mRNA also suggests that its occurrence by itself in the 3′UTR of mRNAs does not make them paraspeckle targets. As already mentioned, the same conclusion can be drawn for IRAlu sequences. In any case, this 15-nucleotide sequence is shown here to be present twice more frequently than a random sequence of the same length in the 3′UTR of all mRNAs. This high prevalence could support an important role of this sequence in RNA biology.

The 11 proteins that were identified as preferentially bound to the 15-nucleotide sequence are, as expected, RNA binding proteins, and among them, six have been previously described as paraspeckle protein components [18]. Among them, HRNPK, as identified by GraphProt [24], displays in its binding peaks a pyrimidine-rich sequence consistent with the known preferences of HNRNPK for C-rich sequences [22]. This appears in accordance with the 15-nucleotide sequence we identified here, in which more than half of the nucleotides are pyrimidines. Moreover, the three KH RNA-binding domains of HNRNPK were previously shown to act cooperatively in binding sequences with triplets of C/T-rich regions [25], fitting the sequence architecture of the entire 3′UTR Calr mRNA. However, while the presence of multiple cytosine patches appears to be necessary for interaction with HNRNPK, it is not always sufficient and does not guarantee an interaction with HNRNPK [26]. This is illustrated here by the inability of the central 3′UTR region of Calr mRNA (3′UTR-C2 fragment) to associate with paraspeckles despite the presence of numerous cytosine patches (see Appendix A). It has also been suggested that the C/T-rich motif is preferentially bound in a structured context. The interplay of C-patch spacing and secondary structure formation influences HNRNPK RNA recognition since HNRNPK recognition of C-patches depends on positioning within the RNA structure [27]. Interestingly, predictive structural analysis of the 30-bases oligonucleotide we used in the binding experiments demonstrates the accessibility of C-patches located both within the 15-nucleotide sequence identified here and in its 5′ adjacent region (Appendix A). It is then tempting to speculate that the HNRNPK binding to the 15-nucleotide sequence is potentiated by the adjacent 5′ C-rich region. This could also be the case for the ability of the other five paraspeckle proteins identified here to bind to the 15-nucleotide sequence. A 42 nucleotides fragment containing three stretches of at least six pyrimidines (C/T), which has been named SIRLOIN (SINE-derived nuclear RNA LOcalizatIoN), has been previously shown to bind HNRNPK and direct the nuclear enrichment of mainly lncRNAs but also certain mRNAs [28]. As suggested by these authors and in view of our present results, multiple independent pathways are likely responsible for nuclear enrichment in lncRNAs and mRNAs, recognizing specific RNA sequences and/or other features of the ribonucleoproteins.

## 4. Materials and Methods

### 4.1. Cell Line Culture and Preparation of Stably-Transfected Cell Lines

GH4C1 cells, a rat pituitary somatolactotroph cell line, were obtained in 2012 from ATCC (CCL-82.2, lot number: 58945448, Molsheim, France) with certificate analysis and were confirmed to be free of mycoplasma (MycoAlert, Lonza, Levallois-Perret, France). They were grown in HamF10 medium supplemented with 15% horse serum and 2% fetal calf serum. GH4C1 cells were synchronized between themselves by a replacement of fresh medium. For the generation of stable GH4C1 cell lines, cells were transfected with plasmid constructs expressing a neomycin resistance gene by Lipofectamine 3000 (Invitrogen, Cergy Pontoise, France). Cells were selected with 250 mg/mL G418 (Invitrogen) beginning 48 h after transfection.

### 4.2. Plasmid Constructs

The 3′UTR of Calr (3′UTR-Calr) mRNA (574 bases) was amplified using classic 3′ RACE by dual PCR from total RNA obtained from GH4C1 cells (see Appendix A for primers). The sequence was then inserted in the plasmid pEGFP-C1 (Clontech, Mountain View, CA, USA) at the Sma1 site. Using this 3′UTR construction, PCR was performed in order to obtain three overlapping subsequences named C1, C2, and C3 (see green segments in Appendix A). These sequences were inserted using the SmaI site in pEGFP-C1. Primers (IDT, Coralville, IA, USA) used for these constructions were listed in Appendix A. All plasmid constructs were verified by sequencing (Genewiz, Leipzig, Germany).

A 30-mer specific oligonucleotide named 30S (Appendix A) corresponding to the position 459 to 488 of the 3′UTR-Calr mRNA (see Appendix A), a 30S probe with mutation in position 462–464 (CCC replaced by ATA, 30SSM Appendix A), and a 30-mer non-specific oligonucleotide named 30NS (Appendix A) corresponding to 30S mutated in position 462–464 (CCC replaced by ATA) and in position 466–480 (every odd base replaced by its complementary inverse base) were purchased from IDT. 30S and 30NS oligonucleotides were inserted in the plasmid pEGFP-C1 (Clontech) at BglII and Kpn1 sites.

### 4.3. Mutagenesis

The reporter plasmid pEGFP-C1 containing the entire 3′UTR-Calr mRNA was mutated in the 15-nucleotide sequence present in position 37–51 (Mut1) or in position 466–480 (Mut2) or in both positions (Mut1/2) by inverse PCR using the oligonucleotides listed in Appendix A. The mutant cDNA sequences were controlled by sequencing (Genewiz).

### 4.4. RNA Expression Analysis

From GH4C1 cells, nuclear and cytoplasmic RNA were prepared using the Nucleospin RNA XS kit (Macherey Nagel, Hoerdt, France), and total RNA was prepared using the Nucleospin RNA kit (Macherey Nagel). Nuclear and cytoplasmic RNA isolation was performed using 10 cm cell dishes that were rinsed twice with ice-cold PBS and incubated in 1 mL of ice-cold cell lysis buffer A (10 mmol/L Tris pH 7.4, 3 mmol/L MgCl_2_, 10 mmol/L NaCl, and 0.5% NP-40). Nuclei and cytoplasm were separated by centrifugation (500× *g* for 5 min). One-sixth of the supernatant was used to prepare cytoplasmic RNA. To obtain pure nuclear RNA, the nuclear pellets were subjected to two additional washes with 1 mL lysis buffer A and were then extracted with Nucleospin RNA XS kit reagent (Macherey Nagel).

The lack of cross-contamination between nuclear and cytoplasmic fractions was controlled by Western blotting experiments with antibodies directed against a nuclear protein, activating transcription factor 2 (ATF2) (anti-ATF2, sc-187, Santa Cruz Biotechnology, Heidelberg, Germany), on the one hand, and a cytoplasmic protein, mitogen-activated protein kinase kinase 1/2 (MEK1/2) (anti-MEK1/2, Cell Signaling, Leiden, The Netherlands), on the other hand.

RNA (500 ng) was then used for cDNA synthesis performed with a High Capacity RNA to cDNA kit (Applied Biosystems, Courtaboeuf, France). Real-time qPCR was performed on a 7500 fast Real-Time qPCR system (Applied Biosystems) using Fast SYBR Green mix (Applied Biosystems). The primer sequences used in qPCR are given in Appendix A. mRNA accumulation given as the relative ratio of nuclear versus cytoplasmic mRNA was normalized relative to Gapdh mRNA levels.

### 4.5. RNA Immunoprecipitation (RIP) Experiments

GH4C1 cells grown in 10 cm dishes were rinsed two times with 5 mL cold phosphate buffer saline (PBS). Cells were then harvested by scraping in ice-cold PBS and transferred to a centrifuge tube. After centrifugation (2500× *g* for 5 min), cells were pelleted and suspended in 100 μL of Polysome lysis buffer (PLB; 10 mM HEPES, pH 7.0, 0.1 M KCl, 5 mM MgCl_2_, 0.5% NP40, 1 mM DTT, 100 U/mL RNAse OUT, and complete protease inhibitor cocktail). After mixing by pipetting up and down, cells were kept on ice for 5 min to allow the hypotonic PLB buffer to swell the cells. The cell lysate was then aliquoted and stored at −80 °C.

Cell lysate was centrifuged at 14,000× *g* for 10 min at 4 °C and diluted 1/100 in NET2 buffer (NET2 buffer corresponded to NT2 buffer: 50 mM Tris-HCl, pH 7.4, 150 mM NaCl, 1 mM MgCl_2_, and 0.05% NP40 added with 1 mM DTT, 20 mM EDTA, and 200 U/mL RNAse Out). An aliquot of diluted cell lysate was removed and represented the starting material, or ’input’, which was processed alongside the immunoprecipitation to compare with immunoprecipitated mRNAs at the end. RIP experiments were performed overnight at 4 °C on diluted cell lysate with antibodies to NONO (ab45359, Abcam, Paris, France), SFPQ (ab38148, Abcam), PSPC1 (SAB4200068, Sigma-Aldrich, Saint-Quentin Fallavier, France), and RBM14 (ab70636, Abcam) or non-specific rabbit polyclonal antibody (anti-Furin, sc-20801). After incubation was completed, 15 μL of Magna ChIP protein A magnetic beads (16–661, Millipore, Molsheim, France) were added for 1 h at 4 °C. Beads were washed 6 times with cold NT2 buffer and treated with proteinase K for 30 min at 55 °C. RNA eluted from beads was purified using Nucleospin RNA XS (Macherey-Nagel, Hoerdt, France) and processed for cDNA synthesis using a High Capacity RNA to cDNA kit (Applied Biosystems, Courtaboeuf, France).

### 4.6. Neat1 RNA Pull-Down

Neat1 RNA pull-down is a hybridization-based strategy we previously published [10] that uses complementary oligonucleotides to purify Neat1 RNA together with its targets from reversibly cross-linked extracts. Two biotinylated antisense DNA oligonucleotide probes that target accessible regions of the lncRNA Neat1 were used for Neat1 RNA specific pull-down (SO1 and SO2), and one biotinylated irrelevant probe (NSO) was used for Neat1 RNA non-specific pull-down (Appendix A). All three probes were biotinylated at the 3′ end.

Briefly, 10 cm cell dishes were incubated in 1 mL of ice-cold cell lysis buffer as described above. Nuclei were scraped and separated by centrifugation (500× *g* for 5 min). The nuclear pellets were then fixed with 1% paraformaldehyde in PBS for 10 min at room temperature. Crosslinking was then quenched with 1.25 M glycine for 5 min. Cross-linked nuclei were rinsed two times again with PBS and pelleted at 500 g. The nuclear pellets were stored at −80°C. To prepare lysates, nuclear pellets were suspended in lysis buffer (50 mM Tris, pH 7.0, 10 mM EDTA, 1% SDS added with a protease inhibitor cocktail and RNAse-Out) and sonicated using BioruptorPlus (Diagenode, Seraing, Belgium) by 2 pulses of 30 s, allowing complete lysate solubilization. RNA was in the size range of 400 to 2000 nucleotides. Nuclear lysates were diluted *v*/*v* in hybridization buffer (750 mM NaCl, 1% SDS, 50 mM Tris, pH 7.0, 1 mM EDTA, 15% formamide). The two specific or the non-specific probes (100 pmol) were added to 1.2 mL of diluted lysate, which was mixed by end-to-end rotation at room temperature for 4 to 6 h. Streptavidin-magnetic C1 beads (Dynabeads MyOne Streptavidin C1–Invitrogen Life Technologies) were added to the hybridization reaction (50 μL per 100 pmol of probes), and the whole reaction was mixed overnight at room temperature. Beads–biotin-probes–RNA adducts were captured by magnets (Invitrogen) and washed five times with a wash buffer (2 SSC, 0.5% SDS). After the last wash, the buffer was removed carefully. For RNA elution, beads were suspended in 100 μL RNA proteinase K buffer (100 mM NaCl, 10 mM Tris, pH 7.0, 1 mM EDTA, 0.5% SDS) and 100 μg proteinase K (Ambion, Thermo Fisher, Waltham, MA, USA). After incubation at 45 °C for 45 min, RNA was isolated using NucleoSpin RNA XS (Macherey-Nagel). Eluted RNA was subject to RT–qPCR for the detection of enriched transcripts.

### 4.7. RNA-Protein Pull-Down

50 pmol of a specific 3′ biotin-TEG oligonucleotide specific RNA probe (30S or 30SSM) or non-specific RNA (30NS) probe (Appendix A) (IDT) were incubated for 15–30 min at room temperature under agitation with 50 μL of Streptavidin Magnetic Beads from the PierceTM Magnetic RNA-Protein Pull-Down Kit. 100 μL of a Master Mix containing 15% glycerol and nuclear GH4C1 protein extracts were incubated 30–60 min at 4 °C with agitation. After three washes, protein complexes were eluted from RNA by adding 50 μL of Elution Buffer to the beads and incubating 15–30 min at 37 °C with agitation. Samples were then loaded onto a NuPAGE 4–12% Bis-Tris gel (Invitrogen) for mass spectrometry analysis.

### 4.8. MS Analysis

Samples were allowed to migrate 5 min at 200 V. The gel was then silver-stained and cut into small pieces. Before trypsin digestion, the gel pieces were destained, rinsed, and then reduced with dithiothreitol 20 mM and alkylated with iodoacetamide 10 mM in 100 mM NH_4_HCO_3_.

Samples were then incubated overnight at 37 °C with 12.5 ng/μL trypsin (sequencing grade; Promega, Charbonnières-les-Bains, France) in 25 mM NH_4_HCO_3_. Gel pieces were then extracted 3 times with 50% acetonitrile and 0.1% formic acid and evaporated to dryness using a speed vac.

The dried samples were resuspended in 8 μL of 2% acetonitrile and 0.1% formic acid and analyzed by mass spectrometry using a hybrid Q-Orbitrap mass spectrometer (Q-Exactive, Thermo Fisher Scientific, Waltham, MA, USA) coupled to a nanoliquid chromatography (LC) Dionex RSLC Ultimate 3000 system (Thermo Fisher Scientific, USA). Peptide separation was performed on an Acclaim PepMap RSLC capillary column (75 μm × 15 cm, nanoViper C18, 2 μm, 100 Å) at a flow rate of 300 nl/min. Solvent A was 0.1% formic acid. The analytical gradient was run with various percentages of solvent B (acetonitrile with 0.1% formic acid) in the following manner: (1) 2.5–25% for 57 min, (2) 25–50% for 6 min, (3) 50–90% for 1 min, and (4) 90% for 10 min. MS spectra were acquired at a resolution of 35,000 within a mass range of 400–1800 *m*/*z*. Fragmentation spectra of the 10 most abundant peaks (Top10 method) were acquired with high-energy collision dissociation (HCD).

### 4.9. Bioinformatics

#### 4.9.1. Mass Spectrometry Data Analysis

All mass spectrometry RAW files were uploaded into MaxQuant version 1.5.1.0 [29] and searched against a rat SwissProt protein database (December 2018 release). The following parameters were used for the search: trypsin/P enzyme with up to 3 missed cleavages allowed; carbamidomethylation of cysteine was set to fixed modification; with oxidation of methionine, N-terminal protein acetylation was set as variable modifications; the first search peptide tolerance was set to 20 ppm against a small ‘mouse-first-search’ database for the purpose of mass recalibration, and the main search was performed at 4.5 ppm; contaminants were included in the search; the database was reversed for the purpose of calculating the peptide and protein level false-discovery rate (FDR) at 1%.

For statistical comparison, from two independent experiments, two datasets of 3 replicates of 30S and 3 replicates of 30SSM versus 3 replicates of 30NS were analyzed with R (version 4.4.1) after importation with the PerseusR package (version 0.3.4). Contaminant proteins, proteins only identified by site and reverse identifications, were filtered out of the dataset, and the ‘Majority protein IDs identified by at least two peptides were kept for the analysis.

#### 4.9.2. Nucleotide Sequence Analysis

This analysis was conducted using the bioconductor Biostrings package (v 2.76) of R (v 4.5) software.

Consensus motif

Local alignment of the 3′UTR-C1 and the 3′UTR-C3 fragments was performed using the pairwiseAlignment function.

3′UTR sequences

The rat DNA sequences and the coordinates of the 3′UTR were obtained from the m RatBN7.2 dna sequence and rattus_norvegicus/Rattus_norvegicus.mRatBN7.2.112.gtf in Ensembl, respectively.

For each gene, only the sequence of the longest 3′UTR described, when available, was taken into account, leading to a list of 19 290 sequences, which was considered as the background file. From this file, a subset of sequences was extracted, those belonging to the 3928 Neat1 RNA targets we previously established (11), leading to a list of 3347 sequences considered as the specific file.

Consensus matrix

The specific and the background files were scanned for the TGCCYCCAGGRCTGG motif using the vmatchPattern function, allowing 4 mismatches.

A consensus matrix was generated from the 2190 sequences obtained, using the consensusMatrix function.

A logo was drawn from this matrix using the seqLogo package (v 1.70) from bioconductor.

### 4.10. RNA-FISH

To detect Neat1 or/and Calr RNA, GH4C1 cells grown on glass coverslips coated with poly-ornithine were fixed in 3.6% formaldehyde. Hybridization was carried out using Custom Stellaris FISH Probes (Biosearch Technologies, Novato, CA, USA). Probes used for dual FISH are Calr probes labeled with Quasar 670 Dye and Neat1 probes labeled with Quasar 570 Dye. Nuclei of the cells were counterstained by Hoechst solution (1 μmol/mL).

### 4.11. Cosinor and Statistical Analysis

Cosinor and statistical analyses were performed using Prism4 software (GraphPad Software, Inc., San Diego, CA, USA).

For cosinor analysis, mean experimental values (±SEM), expressed as a percent of initial value, were fitted using Prism4 software (GraphPad Software, Inc.) by a non-linear sine wave equation: Y = Baseline + Amplitude × sin (Frequency × X + Phase-shift), where Frequency = 2pi/period and period = 24 h. Goodness of fit was quantified using R-squared, experimental values being considered well fitted by cosinor regression when the R-squared was higher than 0.55. A statistically significant circadian oscillation was considered if the 95% confidence interval for the amplitude did not include the zero value (zero-amplitude test) [30,31].

One-way ANOVA followed by Fisher’s LSD test was used to test for significant differences in mean between groups.

A Chi-square test was used for comparing two proportions (R prop.test)

Values were considered significantly different for *p* < 0.05 (*), *p* < 0.01 (**) or *p* < 0.001 (***).

## 5. Conclusions

The present results allow the 2190 sequences found in the 3′UTR of paraspeckle mRNA targets from a pituitary cell line to design a motif and to propose a novel matrix in the 3′UTR of mRNAs that may contribute to their nuclear retention through association with paraspeckles. Furthermore, these results suggest that the recognition of this sequence in a structured RNA context is important to bridge paraspeckles with their mRNA targets. However, the sequence we described here is widely present in the 3′UTR of mRNAs, whether they are or they are not paraspeckle targets, and it thus may be assumed that other specific RNA sequences and/or other associated proteins are involved in mRNA paraspeckle nuclear retention. This underlines that the description of a simple motif recognition site to predict complex binding events may be too simplistic. Nevertheless, based on these findings, the novel matrix proposed can be presented as a promising candidate for the elusive recruiting elements of mRNAs by paraspeckles and lay the groundwork for future studies to interrogate the role of cooperative other sequences in paraspeckle mRNA nuclear retention.

## Figures and Tables

**Figure 1 ijms-26-06488-f001:**
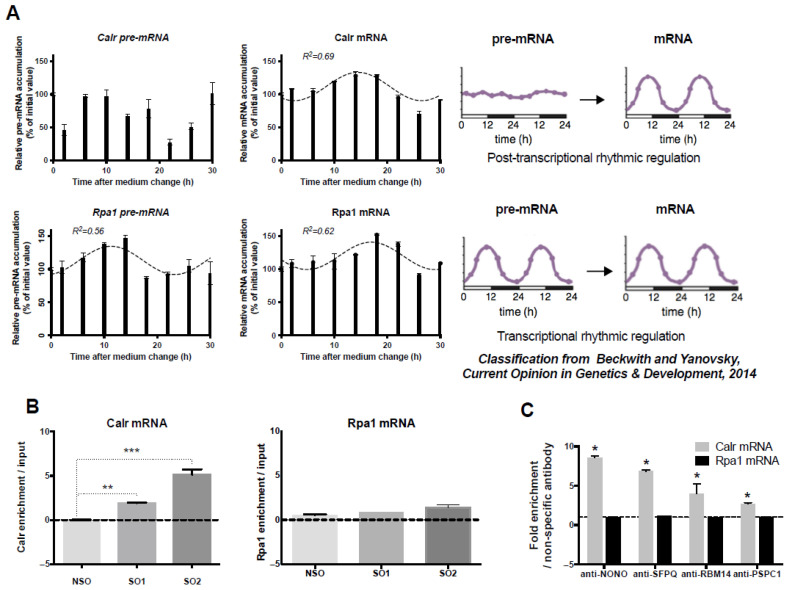
Post-transcriptional control of Calr mRNA circadian expression and association of Calr mRNA with paraspeckle components; (**A**) Post-transcriptional control of Calr mRNA circadian expression: The expression of Calr pre-mRNA and Calr mRNA is determined by RT-qPCR over a 30 h time period. Experimental values of Calr pre-mRNA expressed as a percent of the initial value obtained at T = 0 (Time of medium change) cannot be adequately fitted (R^2^ < 0.55) with a non-linear sine wave equation in which the period value is set to 24 h. By contrast, experimental values of Calr mRNA expressed as a percent of the initial value obtained at T = 0 can be adequately fitted (R^2^ > 0.55) with a non-linear sine wave equation in which the period value is set to 24 h. According to the classification from Beckwith and Yanovsky [16], such a rhythmic mRNA pattern associated with an arrhythmic pre-mRNA pattern corresponds to post-transcriptional rhythmic regulation (Right Panel). At the opposite, the rhythmic expression pattern of Rpa1 pre-mRNA and Rpa1 mRNA gives an example of an mRNA submitted to a transcriptional rhythmic regulation according to the classification from Beckwith and Yanovsky [16] (Right Panel). Indeed, experimental values of both Rpa1 pre-mRNA and Rpa1 mRNA obtained by RT-qPCR over a 30 h time period and expressed as a percent of the initial value obtained at T = 0 (Time of medium change) can be adequately fitted (R^2^ > 0.55) with a non-linear sine wave equation in which the period value is set to 24 h. (**B**) Association of Calr mRNA with Neat1: Enrichment in Calr mRNA relative to input after Neat1 RNA pull-down with two different specific biotinylated oligonucleotides, Specific Oligonucleotide 1 (SO1) and Specific Oligonucleotide 2 (SO2), as compared to a non-specific oligonucleotide (NSO), shows the association of Calr mRNA with Neat1; ** *p* < 0.01 *** *p* < 0.001. At the opposite, the lack of enrichment in Rpa1 mRNA relative to input after Neat1 RNA pull-down with SO1 and SO2 compared to NSO attests that Rpa1 is not associated with Neat1. (**C**) Association of Calr mRNA with four core paraspeckle proteins: Enrichment in Calr mRNA after RNA Immuno-Precipitation (RIP) with antibodies directed against NONO, SFPQ, RBM14, and PSPC1 relative to an irrelevant antibody shows the association of Calr mRNA with the four core paraspeckle proteins; * *p* < 0.05. By contrast, in the same experiments, antibodies directed against NONO, SFPQ, RBM14, and PSPC1 do not provide an enrichment in Rpa1 mRNA as compared to an irrelevant antibody, showing that Rpa1 mRNA is not associated with core paraspeckle proteins.

**Figure 2 ijms-26-06488-f002:**
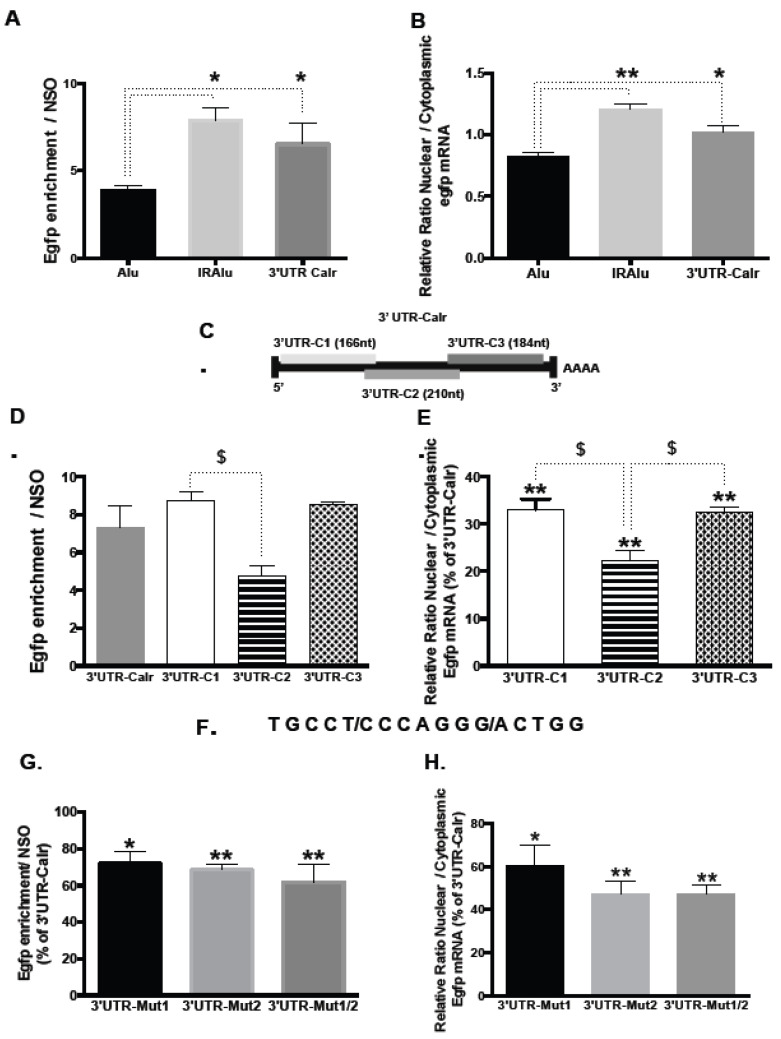
A 15-nucleotide sequence in 3′UTR involved in the nuclear retention of Calr mRNA by paraspeckles; (**A**,**B**) Involvement of 3′UTR in Calr mRNA association with paraspeckles (**A**) Enrichment in Egfp mRNA after Neat1 RNA pull-down with Specific biotinylated Oligonucleotide 2 (SO2) relative to Non-Specific Oligonucleotide (NSO). The relative enrichment in Egfp mRNA obtained after RNA pull-down (n = 6 for each cell line) is statistically different in IRAlu-egfp and 3′UTR-Calr-egfp versus Alu-egfp cell lines but not different between IRAlu-egfp and 3′UTR-Calr—egfp cell lines * *p* < 0.05 (**B**) Nuclear and cytoplasmic Egfp mRNA were quantified by qPCR in Alu-egfp, IRAlu-egfp and 3′UTR-Calr-egfp cell lines and normalized to the relative amount of Gapdh mRNA (n = 3 for each cell line). Ratio of nuclear versus cytoplasmic Egfp mRNA levels is statistically higher in IRAlu-egfp and 3′UTR-Calr-egfp cell lines compared to Alu-egfp cell line but not different between IRAlu-egfp and 3′UTR-Calr-egfp cell lines; * *p* < 0.05 ** *p* < 0.01. (**C**–**E**) Sub-regions of 3′UTR-Calr mRNA associated with paraspeckles (**C**) Schematic representation of the three fragments generated from 3′UTR-Calr mRNA. 3′UTR-Calr mRNA was fragmented into three overlapping parts of 166 (3′UTR-C1), 210 (3′UTR-C2), or 184 (3′UTR-C3) nucleotides, respectively. (**D**) Enrichment in Egfp mRNA after Neat1 RNA pull-down with Specific biotinylated Oligonucleotide 2 (SO2) relative to Non-Specific Oligonucleotide (NSO). The relative enrichment in Egfp mRNA obtained after RNA pull-down (n = 3 for each cell line) does not differ between the entire 3′UTR-Calr- and 3′UTR-C1- or 3′UTR-C3-cell lines but is statistically different between 3′UTR-C1- versus 3′UTR-C2-cell lines; $ *p* < 0.05. (**E**). Ratio of nuclear versus cytoplasmic Egfp mRNA levels obtained in 3′UTR-C1, 3′UTR-C2, and 3′UTR-C3 cell lines are normalized to the relative amount of Gapdh mRNA (n = 3 for each cell line) and expressed as a percent of the ratio in the 3′UTR-Calr cell line. The ratio obtained in the 3′UTR-C1, 3′UTR-C2, and 3′UTR-C3 cell lines are significantly lower than that obtained in the 3′UTR-Calr-egfp cell line; ** *p* < 0.01. However, the ratio obtained in 3′UTR-C2 is significantly lower than those obtained in 3′UTR-C1 and 3′UTR-C3 cell lines; $ *p* < 0.05. (**F**–**H**) Identification of a 15-nucleotide sequence involved in paraspeckle association F. Common sequence of 15 nucleotides found in 3′UTR-C1 and 3′UTR-C3 fragments of the 3′UTR-Calr. (**G**,**H**) The complementary inverse base replaced every odd base in the 15-nucleotide sequence present in position 37–51 (3′UTR-Mut1), in position 466–480 (3′UTR-Mut2), or in both positions (3′UTR-Mut1/2). (**G**) Enrichment in Egfp mRNA after Neat1 RNA pull-down with Specific biotinylated Oligonucleotide 2 (SO2) relative to Non-Specific Oligonucleotide (NSO) obtained in 3′UTR-Mut1-, 3′UTR-Mut2-, and 3′UTR-Mut1/2-cell lines is significantly reduced compared to that obtained in the 3′UTR-Calr cell line; * *p* < 0.05 ** *p* < 0.01 (**H**) Ratio of nuclear versus cytoplasmic Egfp mRNA levels obtained in 3′UTR-Mut1, 3′UTR-Mut2, and 3′UTR-Mut1/2 cell lines are normalized to the relative amount of Gapdh mRNA (n = 3 for each cell line) and expressed as a percent of the ratio in the 3′UTR-Calr cell line. The ratio is significantly reduced in 3′UTR-Mut1, 3′UTR-Mut2, and 3′UTR-Mut1/2 cell lines compared to the ratio in the 3′UTR-Calr cell line. * *p* < 0.05 ** *p* < 0.01.

**Figure 3 ijms-26-06488-f003:**
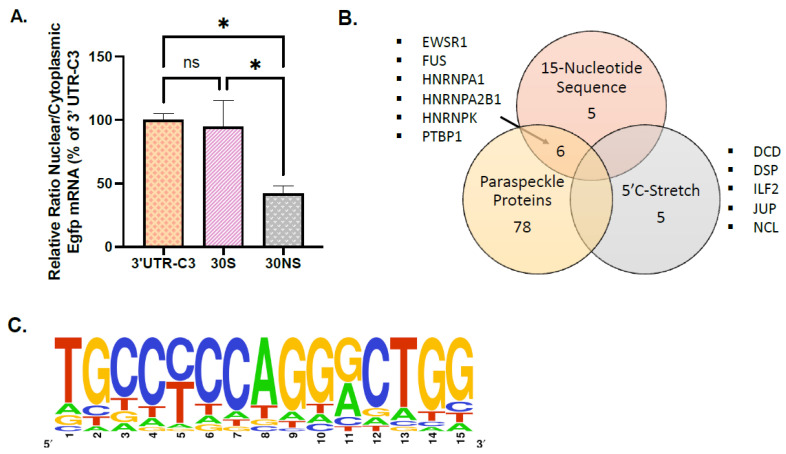
Analysis of proteins associated with the 15-nucleotide sequence identified in 3′UTR-Calr mRNA. (**A**) Ratio of nuclear versus cytoplasmic Egfp mRNA levels obtained in 30S and 30NS oligonucleotide cell lines are normalized to the relative amount of Gapdh mRNA (n = 3 for each cell line) and expressed as a percent of the ratio in the 3′UTR-C3 cell line. Ratio of nuclear versus cytoplasmic Egfp mRNA levels is 4-fold statistically higher in 3′UTR-C3- and 30S-egfp cell lines compared to the 30NS-egfp cell line but not different between the 3′UTR-C3- and 30S-egfp cell lines; * *p* < 0.05. (**B**) Mass spectrometry identification of 16 proteins specifically bound to a 30S oligonucleotide containing the 15-nucleotide sequence. Among the 11 proteins (ALF-C1, EWSR1, FUS, HNRNPA0, HNRNPA1, HNRNPA2B1, HNRNPAB, HNRNPD, HNRNPK, PTBP1, and YBX3) that were identified using the 30S and 30SSM probes and not with the 30NS probe, 6 proteins are included in the list of paraspeckle proteins. Five proteins (DCD, DSP, ILF2, JUP, and NCL) identified with the 30S but not with the 30SSM and the 30NS are merely associated with the stretch of C present in 5′ of the 15-nucleotide sequence. None of these 5 proteins are paraspeckle proteins. (**C**) Sequence logo that can be designed from the 2190 sequences found in the 3′UTR of mRNA paraspeckle targets.

**Table 1 ijms-26-06488-t001:** Prevalence of the 15 nucleotides sequence in the 3′UTR of Neat1 RNA targets and in the 3′UTR of all rat RNAs.

	Number of 3′UTR	Number of Sequences	Number of Genes	Number of Sequences/Genes
Neat1 RNA targets	3347	2190	1330	1.65
Background	19,290	10,575	6386	1.65
(total RNAs)				

## Data Availability

Raw data will be available upon request.

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
