# Peer review of "Nuclear Retention of mRNAs Through Paraspeckle Protein Binding to a Sequence Determinant in 3′UTR"

_ijms, 2025, doi:10.3390/ijms26136488_

Round 1

Reviewer 1 Report

Comments and Suggestions for Authors

Comments IJMS 2025

Paraspeckles are membrane-less nuclear bodies that show properties of biomolecular condensates. Although their biological role is not entirely clear, they have been shown to bind a number of mRNA species. Association with paraspeckles leads to accumulation of mRNA in the nucleus, thereby adversely affecting their nucleocytoplasmic transport and translation. Paraspeckle mediated nuclear retention of mRNA may be crucial in regulating expression of genes coding for these mRNAs. Why only certain species of mRNA associate with paraspeckles, however, is not entirely clear. Paraspeckle associated mRNAs often have Alu I inverted repeats in their 3’UTR, but not all mRNA species bound to paraspeckles harbor Alu I inverted repeats. Clearly, there are other factors that regulate paraspeckle-mediated nuclear retention of mRNA. The manuscript by Jacq et al., investigated structural features of mRNA that facilitate their association with paraspeckles. Uaing calreticulin (Calr) mRNA, authors showed that the association of Calr mRNA with paraspeckles involves 15-nucleotide tandem repeats in the 3’UTR of mRNA. The 15-nt repeats are preceded by a stretch of C nucleotides in the upstream region. Authors showed that the 15-nt repeat along with the C-rich stretch play a crucial role in nuclear retention of Calr mRNA by paraspeckles. They further demonstrated that the 15-nt repeat bound several paraspeckle specific proteins. Further analysis revealed that 15-nt repeat was present in at least 40% of paraspeckle-associated mRNA in rat pituitary cell line. These observations are novel and compelling and open up avenues for further research in paraspeckle-mediated regulation of gene expressions, especially those linked to circadian rhythms. The following suggestions may help improve the manuscript:

  1. At multiple places in the manuscript, authors have used the term ‘significant enrichment’ of signal or ‘significantly higher signal’ over control/input. Authors may calculate the fold enrichment or fold change in signal and write the numerical value rather than using these vague terms. They then can indicate that the enrichment was statistically significant.
  2. In line 392, authors mention that nuclear and cytoplasmic fractions were authenticated using marker proteins ATF2 and MEK1/2. These are not the proteins that are normally used as markers for nuclear and cytoplasmic fractions. Please provide the reference here and also include the Western blot in Supplementary Data section.
  3. Authors stated that, “…It then appeared that near 40% of the Neat1 RNA targets had at least one identified 15-nucleotide sequence in their 3’UTR…”. Authors may perform ontological analysis to determine if genes that code for these 40% of the Neat1 RNA targets belong to a particular class/category or share some common structural or functional features.
  4. Redundancies between ‘Results’ and ‘Discussion’ sections must be removed to improve the impact of the work.
  5. The typos and grammatical errors at a few places in the manuscript must be corrected.
Comments on the Quality of English Language

It is a well written manuscript. The typos and grammatical errors at a few places in the manuscript must be corrected.

Author Response

Comments 1:

At multiple places in the manuscript, authors have used the term ‘significant enrichment’ of signal or ‘significantly higher signal’ over control/input. Authors may calculate the fold enrichment or fold change in signal and write the numerical value rather than using these vague terms. They then can indicate that the enrichment was statistically significant.

Response 1: Thank you for pointing this out. We have taken this into account and calculate the enrichments obtained; these changes appear in red in the text.

Comments 2:

In line 392, authors mention that nuclear and cytoplasmic fractions were authenticated using marker proteins ATF2 and MEK1/2. These are not the proteins that are normally used as markers for nuclear and cytoplasmic fractions. Please provide the reference here and also include the Western blot in Supplementary Data section.

Response 2:

We have used these protein markers in previous publications to control our nuclear and cytoplasmic fractions. In this revised version of the manuscript, we provide the reference of one of our articles in which we used these protein markers and we have illustrated in Supplementary figure 3 the western blot of these controls.

Comments 3:

Authors stated that, “…It then appeared that near 40% of the Neat1 RNA targets had at least one identified 15-nucleotide sequence in their 3’UTR…”. Authors may perform ontological analysis to determine if genes that code for these 40% of the Neat1 RNA targets belong to a particular class/category or share some common structural or functional features.

Response 3:

As suggested by the reviewer, a Panther ontology analysis was performed on the 40% of the Neat1 RNA targets with the 15-nucleotide sequence.This analysis is now given in Supplemental table 3.

Comments 4:

Redundancies between ‘Results’ and ‘Discussion’ sections must be removed to improve the impact of the work.

Response 4:

Thank you for pointing this out. As suggested by the reviewer, some parts of the discussion (in red) have been rewritten so as not to paraphrase the results

Comments 5:

The typos and grammatical errors at a few places in the manuscript must be corrected.

Response 5:

Special attention was paid to typography and typing errors during proofreading

4. Response to Comments on the Quality of English Language

Point 1: The English could be improved to more clearly express the research.

Response 1: (in red)

We used our internal editing service and corrected the English

Reviewer 2 Report

Comments and Suggestions for Authors

The authors identified a sequence motif in the 3′UTR of Calr mRNA that is required for its enrichment in paraspeckles. Furthermore, they successfully identified potential RNA-binding proteins that can bind this motif. Since these proteins are localized to paraspeckles, they may serve as a bridge between the motif-containing RNA and the paraspeckles. Additionally, the authors suggest that circadian regulation of Calr mRNA expression is exerted at the post-transcriptional level. Taken together, these findings support an interesting hypothesis regarding the physiological function of paraspeckles in circadian rhythm regulation. Although the authors seem cautious about drawing a strong conclusion—likely due to the lack of direct experimental evidence supporting this link—the aim of the manuscript is nevertheless clearly conveyed through the organization of Figure 1 and the other figures in conjunction with the Introduction. The authors also discuss their results carefully in the Discussion section, which is appropriate.

The major concern is that if sequestration of Calr mRNA in paraspeckles is indeed a key mechanism for its circadian regulation, one would expect the mRNA to be predominantly localized in paraspeckles within the nucleus. However, the RNA FISH experiments show that only a small fraction of Calr mRNA overlaps with paraspeckles. This observation appears inconsistent with the biochemical data, raising questions about whether Calr mRNA sequestration in paraspeckles actually occurs to a degree sufficient to drive circadian regulation.
Given this, I am also uncertain about the validity of the experimental system used to identify RNA-binding proteins—specifically, the pull-down assay using synthetic oligonucleotides. At the very least, the authors should consider analyzing available eCLIP data to determine whether it supports the same conclusion.

Moreover, in Figure 1, the authors imply that the circadian regulation of Calr mRNA occurs at the post-transcriptional level, in contrast to Rpa1 mRNA. However, the fluctuation of Calr pre-mRNA appears to precede that of mature mRNA by only one time point, which suggests that it may still be modestly regulated in a circadian manner, even if this is not statistically significant. Given gene-specific differences in transcription rates and RNA stability, I remain unconvinced that Calr expression is primarily regulated post-transcriptionally. That said, this is a subjective impression, and I am not requesting any changes on this point.

Minor point: In line 638, the word "arbor" may be an error and should possibly be "harbor".

Author Response

Thank you very much for taking the time to review this manuscript. Please find the detailed responses below.

Comments 1:

The major concern is that if sequestration of Calr mRNA in paraspeckles is indeed a key mechanism for its circadian regulation, one would expect the mRNA to be predominantly localized in paraspeckles within the nucleus. However, the RNA FISH experiments show that only a small fraction of Calr mRNA overlaps with paraspeckles. This observation appears inconsistent with the biochemical data, raising questions about whether Calr mRNA sequestration in paraspeckles actually occurs to a degree sufficient to drive circadian regulation.

Response 1: Thank you very much for this comment. We agree with the referee that FISH images do not seem to indicate a close association of Calr mRNAs with paraspeckles, but our previous results have shown that destruction of paraspeckles by siRNAs directed against Neat1 or by antisense oligonucleotides results in the complete abolition of the circadian rhythm of calr mRNA which makes it more convincing that Calr mRNA sequestration by paraspeckles may underlie Calr mRNA circadian rhythmicity..

Comments 2:

Given this, I am also uncertain about the validity of the experimental system used to identify RNA-binding proteins—specifically, the pull-down assay using synthetic oligonucleotides. At the very least, the authors should consider analyzing available eCLIP data to determine whether it supports the same conclusion.

Response 2:

The pull-down assay coupled with mass spectrometry identification of proteins binding to an oligonucleotide containing our sequence of interest offers the advantage of being unbiased with respect to the identity of the RNA-binding proteins, in contrast to eCLIP-based approaches, which are designed to test whether a specific protein interacts with a given RNA. Using our strategy, we successfully identified 11 proteins—an outcome that would have required considerably more time and resources if performed using eCLIP.

Comments 3:

Moreover, in Figure 1, the authors imply that the circadian regulation of Calr mRNA occurs at the post-transcriptional level, in contrast to Rpa1 mRNA. However, the fluctuation of Calr pre-mRNA appears to precede that of mature mRNA by only one time point, which suggests that it may still be modestly regulated in a circadian manner, even if this is not statistically significant. Given gene-specific differences in transcription rates and RNA stability, I remain unconvinced that Calr expression is primarily regulated post-transcriptionally. That said, this is a subjective impression, and I am not requesting any changes on this point.

Response 3:

As demonstrated by Menet et al. (doi: 10.7554/eLife.00011) in the liver, numerous genes whose rhythmicity is regulated at the post-transcriptional level exhibit highly variable transcription that is buffered to produce robust rhythmic mRNA expression. Only a few genes display relatively constant transcription compared to mRNA expression. The expression profiles of Calr pre-mRNA and mRNA reported here are fully consistent with the findings of Menet et al. As highlighted by these authors, post-transcriptional mechanisms involve RNA-binding proteins and microRNAs that contribute to the regulation of RNA stability, 3' end processing, and nuclear export.

Comments 4:

Minor point: In line 638, the word "arbor" may be an error and should possibly be "harbor".

Response 4:

Thank you for pointing this error. The word was corrected in the new version of the manuscript.

4. Response to Comments on the Quality of English Language

Point 1: The English is fine and does not require any improvement.